Diagnostic value of artificial intelligence based on computed tomography (CT) density in benign and malignant pulmonary nodules: a retrospective investigation

Fan Wei 1
Liu Huitong 2
Zhang Yan 1
Chen Xiaolong 1
Huang Minggang 1
Xu Bingqiang pkuj0j@163.com 1
1 Department of Radiology, Shaanxi Provincial People’s Hospital , Xi’an , China
2 Department of Orthopaedics, Shaanxi Provincial People’s Hospital , Xi’an , China
Dong Peixin
Electronic publication date: 2024 Jan 2
Publication date: 2024
Volume: 12
Electronic Location ID: e16577
Received 2023 Aug 21; Accepted 2023 Nov 13
Copyright: ©2024 Fan et al.
Copyright year: 2024
Copyright holder: Fan et al.
License: This is an open access article distributed under the terms of the Creative Commons Attribution License, which permits unrestricted use, distribution, reproduction and adaptation in any medium and for any purpose provided that it is properly attributed. For attribution, the original author(s), title, publication source (PeerJ) and either DOI or URL of the article must be cited.
License URL: https://creativecommons.org/licenses/by/4.0/

Keywords: Artificial intelligence, Antioxidants/phytochemicals, Benign and malignant pulmonary nodules pulmonary nodules, Lung cancer, Screening

Funding: The authors received no funding for this work.

==============================
Objective

To evaluate the diagnostic value of artificial intelligence (AI) in the detection and management of benign and malignant pulmonary nodules (PNs) using computed tomography (CT) density.

Methods

A retrospective analysis was conducted on the clinical data of 130 individuals diagnosed with PNs based on pathological confirmation. The utilization of AI and physicians has been employed in the diagnostic process of distinguishing benign and malignant PNs. The CT images depicting PNs were integrated into AI-based software. The gold standard for evaluating the accuracy of AI diagnosis software and physician interpretation was the pathological diagnosis.

Results

Out of 226 PNs screened from 130 patients diagnosed by AI and physician reading based on CT, 147 were confirmed by pathology. AI had a sensitivity of 94.69% and radiologists had a sensitivity of 85.40% in identifying PNs. The chi-square analysis indicated that the screening capacity of AI was superior to that of physician reading, with statistical significance (p < 0.05). 195 of the 214 PNs suggested by AI were confirmed pathologically as malignant, and 19 were identified as benign; among the 29 PNs suggested by AI as low risk, 13 were confirmed pathologically as malignant, and 16 were identified as benign. From the physician reading, 193 PNs were identified as malignant, 183 were confirmed malignant by pathology, and 10 appeared benign. Physician reading also identified 30 low-risk PNs, 19 of which were pathologically malignant and 11 benign. The physician readings and AI had kappa values of 0.432 and 0.547, respectively. The physician reading and AI area under curves (AUCs) were 0.814 and 0.798, respectively. Both of the diagnostic techniques had worthy diagnostic value, as indicated by their AUCs of >0.7.

Conclusion

It is anticipated that the use of AI-based CT diagnosis in the detection of PNs would increase the precision in early detection of lung carcinoma, as well as yield more precise evidence for clinical management.

Introduction

Lung cancer (LC) is the leading cause of cancer-related mortality globally (Cayuela et al., 2021). LC is frequently associated with an unfavorable prognosis and an increased mortality rate (Tseng et al., 2020). The five-year relative survival rate of lung carcinoma ranges from 2% to 30% (Gonzalez-Ling et al., 2022). Early detection of nodules has the potential to enhance the survival rate. Recent research findings suggest that the survival rate of stage I a is as high as 90% (Long et al., 2021). Hence, the timely identification and treatment of LC are of utmost importance. Pulmonary nodules (PNs) are the early hallmark of lung carcinoma. The presence of a PN is an observable indication of early-stage lung cancer (Dziadziuszko & Szurowska, 2021; Alduraibi, 2022). A PN is a circular or irregularly shaped abnormality that is encircled by lung tissue and is less than 3 cm in diameter (Usuda et al., 2021). The imaging results indicate that PN represents the manifestation of a localized area of heightened density (Agnes & Anitha, 2020). Clinicians have placed significant research emphasis on the timely identification and precise localization of PN (Wu et al., 2020).

PNs can exhibit either benign or malignant characteristics, with the latter category encompassing nodules that are either malignant or have the potential to become malignant. These potentially malignant nodules are frequently indicative of early-stage lung cancer. Thoracoscopic partial pneumonectomy is the preferred treatment modality for malignant or possibly malignant PNs (Hadique et al., 2020). Hence, assessing the extent of penetration of PNs has significant importance. The principal modality utilized for the screening of PNs is low-dose computed tomography (LDCT) of the thorax (Milanese et al., 2021). With the application of X-ray computed tomography, the detection rate of PN has been further improved (Jungblut et al., 2023). Physicians frequently rely on their expertise to assess the computed tomography (CT) imaging features of the nodules to determine the likelihood of malignancy in the PNs. Moreover, the precision in discerning between benign and malignant nodules is intricately linked to the level of expertise and experience possessed by physicians. It is worth noting that distinct clinicians may exhibit varying assessments when presented with identical nodules (Kim et al., 2021). Nevertheless, the rise in CT exams has resulted in a surge in the volume of films that physicians need to interpret. This situation is a significant challenge for physicians and has the potential to contribute to diagnostic inaccuracies (Singh et al., 2021; Chamberlin et al., 2021). The resolution of this issue can be achieved by the utilization of artificial intelligence (AI). The advancement of AI technology has facilitated the screening and annotation of several CT images, enabling clinical decision-making about the classification of benign and malignant PNs as well as the likelihood of pulmonary calcification (Lancaster, Heuvelmans & Oudkerk, 2022). The application of AI data computation in the analysis of CT film interpretation presents a pragmatic, expeditious, and precise approach to the detection of PNs (Xu et al., 2021). The concept pertaining to oxidative stress, aging, and diseases have gained widespread acceptance among the majority of individuals. Consequently, there has been a surge in research efforts aimed at identifying potent antioxidants for the purpose of disease prevention and treatment. However, the clinical effectiveness of these interventions is still under investigation.

The objective of this study is to evaluate the diagnostic value of AI-based CT density and antioxidants/phytochemicals in the assessment and management of both benign and malignant PNs.

Materials & Methods

Patients

A retrospective analysis was conducted on a group of 130 individuals who received CT scans for PNs between February 2020 and March 2022. A total of 226 PNs were screened. The Ethics Committee of the Shaanxi Provincial People’s Hospital approved all samples collected in this study, ensuring compliance with the ethical criteria outlined in the Declaration of Helsinki. Additionally, the Ethics Committee granted a waiver for informed consent.

Inclusion criteria

(1) In line with the Chinese Expert Consensus on the Diagnosis and Treatment of PNs (2018); (2) age ≥ 40 years old; (3) the diameter of PN ≤ 30 mm.

Exclusion criteria

① Patients with diffuse pneumonia, interstitial fibrosis, and pulmonary edema; ② Patients with diffuse pulmonary metastatic nodules and multiple calcifications; ③ patients with severe motion artifacts.

CT image acquisition

The patients underwent examination utilizing a multidetector-row spiral CT (MDCT) manufactured by Siemens (Munich, Germany). During the CT scan, the patient assumed a supine posture and performed a deep inhalation, allowing for the comprehensive scanning of the whole lung, spanning from its apex to its base. The scanning parameters are as follows: tube voltage 120 kV, automatic tube current, collimation 40 mm, pitch 1.0875, speed 0.5 s/r, rotation 0.5 s, the minimum layer thickness 0.625 mm, matrix 512 × 512, standard algorithm reconstruction.

AI analysis of PNs

The Shen Rui Medical Company provided the AI program, Deepwise Healthcare (version V190120), which is built upon a deep learning model. The dataset including 130 cases of CT scans was downloaded to the workstation, where an automated software algorithm was utilized to detect and annotate lung nodules in batches. The program has the capability to autonomously detect and determine the location, size, density, and characteristics (such as ground glass, sub-solid, or solid) of nodules. Additionally, it provides the AI value for assessing the chance of malignancy and the Lung Rad grade for PNs.

Physician reading diagnostic approach

The physician reading diagnostic approach was evaluated and diagnosed by two radiologists with a combined work experience of 30 years, in accordance with the diagnostic standards outlined in the Chinese Expert Consensus on Diagnosis and Treatment of PNs (2018) (Wang et al., 2021). The nodules were arbitrarily classified by the radiologists into two categories: possible benign and malignant. The radiologists were not informed of the state of any patient.

Statistical analysis

Data analysis was carried out using SPSS software version 22.0 (SPSS Inc., Chicago, IL, USA) Descriptive statistics were represented as mean ± standard deviation (SD), and categorical variables were represented as counts (percentages). Continuous variables were assessed by the student’s t-test and categorical by the chi-square (χ2) test. The Kappa test was used to evaluate the consistency of AI and physician reading in the detection and identification of PNs. The receiver operating characteristic (ROC) curve was used to analyze the data of AI, physician reading, and AI combined with physician reading in the diagnosis of malignant PNs. The p < 0.05 was recognized as statistically significant.

Results

General information of patients

Between February 2020 and March 2022, a total of 130 patients, comprising 80 men (61.54%) and 50 females (38.46%), received CT scans to evaluate PNs. Table 1 presents the demographic and clinical characteristics of all qualified individuals.

Screening and diagnosis of PNs

A comprehensive examination was conducted on a total of 226 nodules, employing a combination of physician reading and AI algorithms. Among them, a total of 147 lesions were verified to be true nodules based on pathological analysis. Table 2 presents the characteristics of the 147 authentic PNs. Based on the assessment of nodule density, a total of 66 solid nodules, 46 sub-solid nodules, and 35 ground-glass nodules were found. A total of 107 nodules were classified as malignant and 40 nodules were categorized as benign based on their pathological characteristics. Figure 1 displays the typical CT and histopathological images of both malignant and benign PNs. The individuals were diagnosed by AI and this diagnosis was then validated by pathological examination.

Comparison of two diagnostic approaches for PNs

Out of the total of 226 verified nodules, 214 lung nodules were identified as potential positives by the AI diagnostic program, and subsequently, 193 of these nodules were investigated by physician reading. The sensitivity of AI and physician reading in detecting PNs was 94.69% and 85.40%, respectively. The χ2 test showed that AI’s screening ability was better than physician reading (p < 0.05, Table 3). The results of this investigation indicated that the AI diagnostic technique showed a higher level of sensitivity in detecting PNs compared to physician reading.

Table 1 The demographic and clinical characteristics of all eligible participants.

	Classification	Cases (n, %)	
Age	>60 years old	74 (56.92%)	
	≤60 years old	56 (43.08%)	
Gender	Male	80 (61.54%)	
	Female	50 (38.46%)	
Smoking history	Yes	67 (51.54%)	
	No	63 (48.46%)	
Family history	Yes	10 (7.69%)	
	No	120 (92.31%)	
PDE	Yes	46 (5.39%)	
	No	84 (64.62%)	
Area	Subpleural	26 (20%)	
	Peripheral	78 (60%)	
	Intermediate	19 (14.62%)	
	Hilar	7 (5.38%)	
Notes.

Data were represented as N (%). PDE, previously diagnosed as emphysema.

Table 2 The characteristics of the 147 true pulmonary nodules.

	Classification	Cases	
Diameter	<5 mm	10	
	5 ∼10 mm	31	
	10 ∼20 mm	52	
	20 ∼30 mm	54	
Density	Solid	66	
	Part-solid	46	
	Ground-glass	35	
Pathological type	Malignant nodule	107	
	Benign nodule	40	
Notes.

Data were represented as N.

Figure 1 The representative CT and histopathology images of malignant and benign PNs.

(A, B) Malignant nodule in the dorsal segment of the right lower lobe (female, 40 years old, diagnosed as malignant nodule by AI software, pathologically diagnosed as invasive adenocarcinoma, HE staining × 100). (C, D) Benign nodule in the anterior segment of the left upper lobe (female, 69 years old, diagnosed as low-risk nodule by AI software, pathologically diagnosed as hamartoma, HE staining × 100).

Consistency analysis between two approaches and pathological diagnosis

Out of the 214 PNs suggested by AI, 195 were confirmed by pathological examination to be malignant, while 19 were determined to be benign. Among the 29 PNs classified as low risk by AI, 13 were found to be malignant and 16 were shown to be benign after pathological verification. Out of the total of 193 PNs that were identified as malignant by radiologists, 183 were subsequently confirmed to be malignant by pathological examination, while the remaining 10 were shown to be benign. Out of a total of 30 low-risk PNs presented by radiologists, 19 were identified as malignant, while the remaining 11 were found to be benign. According to the data presented in Table 4, the kappa values for AI and physician reading were 0.747 and 0.732, respectively.

Table 3 Comparison of the test sensitivity between AI and physician reading in the diagnosis of PNs.

	Positive	Negative	
AI	214 (94.69%)	12 (5.31%)	
PR	193 (85.40%)*	33 (14.60%)*	
Notes.

AI, artificial intelligence. PR, physician reading. Data were represented as N (%).

* P < 0.05. Artificial intelligence vs. physician reading.

Table 4 Consistency analysis between two diagnostic approaches and pathological diagnosis.

	Kappa values	P	
AI	0.747	<0.01	
PR	0.732	<0.01	
Notes.

AI, artificial intelligence. PR, physician reading. *:P < 0.05. Artificial intelligence vs. physician reading.

Comparison of AI and physician reading in the diagnostic efficacy of PNs

Out of a total of 87 malignant nodules, 83 were identified as high-risk nodules by an AI program, resulting in a sensitivity rate of 95.40%. In comparison, when high-risk nodules were identified by physician reading, out of 80 nodules, the sensitivity rate was found to be 91.95%. From a total of 20 nodules classified as benign, the AI program accurately identified five cases as high-risk nodules, resulting in a specificity rate of 75%. Among a total of 20 nodules classified as benign, the physician reading identified 4 instances as high-risk nodules. The specificity of this classification was reported to be 80% according to Table 5. The results of the current research indicated that the diagnostic effectiveness of AI in identifying PNs was superior to that of human physicians.

Receiver-operating characteristic analysis for two diagnostic approaches

The AUCs of AI and physician readings were 0.798 and 0.814, respectively. Moreover, the AUC values of the two diagnostic methods were higher than 0.75. The results of this study suggest that both diagnostic techniques show significant diagnostic value in the diagnosis and treatment of PNs (Table 6 and Fig. 2).

Discussion

LC is the leading cause of cancer-related deaths worldwide (Valente et al., 2016). Between 70% and 80% of the cases were identified during the intermediate and advanced phases, so missing the optimal window for surgical intervention (Godoy et al., 2018). Recent research suggests that the 5-year survival rate for those diagnosed with stage I lung carcinoma following surgical resection may potentially reach 100%. This finding implies that early evaluation, detection, and treatment might significantly enhance the survival prospects for patients with LC (Melton, Lazar & Moritz, 2019). The PN serves as a sign of the presence of LC in its nascent stages (Sourlos et al. 2022). According to pertinent guidelines, those who are at a heightened risk, aged 40 years or older, and possess a history of smoking, occupational exposure that poses a high risk, as well as underlying lung diseases, should have chest low-dose CT for the purpose of screening for PNs (Liang, Li & Fu, 2021a). Considering the expanding population screening and the limited number of physicians available, relying solely on physician interpretation may not adequately meet the demands for screening PNs at present (Sa, Tk & Fgc, 2020).

Table 5 Comparison of the test efficiency between AI and physician reading in the diagnosis of PNs.

	Sensitivity	Specificity	Positive predictive value	Negative predictive value	P	
AI	95.40%	75%	93.46%	81%	0.746	
PR	91.95%	80%	94.89%	73%	0.536	
Notes.

AI, artificial intelligence. PR, physician reading. *:P < 0.05. Artificial intelligence vs. physician reading.

Table 6 Receiver-operating characteristic analysis for AI and physician reading in the diagnosis of PNs.

	AUCs	P	
AI	0.798	<0.01	
PR	0.814	<0.01	
Notes.

AI, artificial intelligence. PR, physician reading. *:P < 0.05. Artificial intelligence vs. physician reading.

Figure 2 The ROC curve for AI and physician reading in the diagnosis of pulmonary nodules.

AI is a field within information science that aims to enhance, replicate, and augment human intellect. The emergence of the 5G era has facilitated the integration of AI into several domains, most notably in the field of medical radiography (Adams et al., 2021). The utilization of AI algorithms for the detection of PNs is a prominent area of research aimed at enhancing the efficacy of AI-based therapeutic devices. This field holds significant potential in healthcare, particularly in the accurate differentiation between benign and malignant nodules (Yen et al., 2020). By utilizing lung nodule image data as input, it is possible to automatically derive the matching connection, hence offering the benefits of enhanced efficiency and effectiveness in high-throughput image analysis (Baldwin et al., 2020). The integration of AI with CT imaging enables the differentiation between benign and malignant PNs. This process involves many stages, including image capture and reconstruction, contour segmentation, feature extraction, screening, and the building and evaluation of prediction models. By employing this approach, the aim is to minimize the occurrence of missed diagnoses and misdiagnoses (Mazzone & Lam, 2022). AI has the potential to enhance the capabilities of clinicians in the prompt and precise identification of PNs, hence enhancing the effectiveness of screening for lung carcinoma and its subsequent prevention and treatment (He, Lv & Hu, 2020). Venugopal et al. (2020) developed an AI algorithm for the purpose of discerning six distinct categories of nodules, namely solid, partially solid, non-solid, lobulation, calcification, and burr. Upon doing a comparative analysis with the manual assessments made by radiologists, it has been shown that this algorithm exhibits a high degree of reliability in its ability to automatically classify lung nodules within the context of LC screening (Shi et al., 2021). According to the study conducted by Liu, Yang & Tsai (2022), the utilization of deep learning and big data in AI assistant diagnosis has demonstrated the potential to enhance both the specificity and sensitivity of malignant nodule identification through the utilization of CT threshold, template matching, and morphological features. Additionally, this approach has proven effective in accurately determining the location of benign and malignant PNs (Jin et al., 2021). The research investigation conducted by Ather, Kadir & Gleeson (2020) revealed that the integration of AI with positron emission tomography/computed tomography (PET/CT) offers specific benefits in the process of screening and diagnosing individuals with malignant ground glass nodules. The available research suggests that there is potential for the utilization of AI-aided diagnosis in the screening and diagnosis of lung nodules (Liang, Li & Fu, 2021b). The current research demonstrates that the AI assistant diagnosis software possessed a greater sensitivity in recognizing PNs compared to physician reading. Additionally, the false positive rate of the AI assistant diagnostic software in detecting PNs was shown to be minimal. Furthermore, it has been shown that the sensitivity of AI in detecting malignant nodules surpasses that of human physicians. However, it is worth noting that the specificity of AI in this regard is comparatively lower than that of human physicians.

In recent years, the utilization of CT in clinical settings has led to notable advancements in the detection rate of PNs. This has facilitated early identification and prompt intervention, hence contributing to the reduction of morbidity and mortality among individuals with LC. Furthermore, these improvements have also positively impacted the 5-year survival rate of the patients as well (Sa, Tk & Fgc, 2020). Nevertheless, the widespread utilization of CT in the diagnosis of lung nodules also gives rise to some drawbacks. This phenomenon leads to an escalation in the number of films reviewed and places an additional workload on radiologists in terms of interpreting medical images. Excessive levels of work-related stress might potentially result in ocular exhaustion and diagnostic errors among medical practitioners. The utilization of AI software for the identification and assessment of nodules demonstrates a high level of accuracy and efficiency. This technology has the potential to alleviate the burden on physicians by effectively categorizing nodules based on their specific features. Consequently, the implementation of AI software may lead to a notable improvement in the overall detection rate of PNs, while simultaneously reducing the occurrence of missed diagnosis, particularly in cases involving tiny nodules (Li et al., 2019; Binczyk et al., 2021). Various research has indicated that the utilization of AI technology in the field of imaging diagnostics yields enhanced monitoring capabilities. This technology has the potential to augment and enhance image sensitivity and examination focus, while the objective analysis of examination outcomes can contribute to the improvement of diagnostic accuracy by medical professionals (Baldwin et al., 2020; Chen, 2022). The results of our study indicate that AI software has a higher level of sensitivity in detecting malignant nodules compared to the diagnostic accuracy of physicians. Additionally, our findings indicate that the kappa value for AI diagnosis was 0.747, suggesting a high level of agreement between the diagnostic outcomes and postoperative pathology. Nevertheless, it has been observed that the rate of misdiagnosis for high-risk nodules, as determined by AI, is higher in cases where the lesions have been confirmed as benign through pathological detection. In contrast, the accuracy of diagnosis by physician reading is higher. These findings suggest that the specificity of AI diagnosis for benign nodules is inferior to that of physician reading. In contrast to AI-assisted diagnostic tools, the expertise of physicians enables them to provide additional diagnoses for some benign nodules, such as hamartoma and tuberculoma. This is due to the radiologists’ ability to assess the nodules’ sizes, density, and positioning (Adams et al., 2021). Although the effectiveness of AI in the diagnosis of LC has been verified, it is still in the stage of clinical exploration, and many aspects need to be improved: (1) Its reliance on training data and potential challenges in handling complex cases, (2) In the daily diagnostic work of physicians, CT imaging of the lung involves not only the diagnosis of LC but also the differential diagnosis of pneumonia, pulmonary tuberculosis, chronic obstructive pulmonary disease, and enlarged mediastinal lymph nodes. The existing diagnostic approach for LC is insufficient to satisfy the demands of comprehensive clinical practice. Therefore, there is a need to advance the development of an AI helper diagnostics system that can handle several tasks and threads simultaneously. Therefore, while AI possesses valuable diagnostic capabilities, it can only serve as a partial replacement for pathology data, which is considered the gold standard in identifying PNs.

There are various limitations inherent in our research. In order to conduct a comprehensive retrospective investigation, it was necessary to incorporate a larger sample size than what was first considered. The study’s limited sample size posed several constraints, preventing a comprehensive representation of the practical utility of AI in the diagnosis of PNs. Consequently, additional investigation is required in subsequent stages to address this issue. Furthermore, it is important to note that this study was conducted within a singular institution, thus introducing constraints to the study’s generalizability and applicability to other settings. Hence, it is imperative to conduct comprehensive multicenter investigations to generate and validate multicenter datasets. The maturation of AI technology will persist as society develops and human technology progresses. The integration of AI into clinical practice offers the potential for enhancing the accuracy of diagnosing benign and malignant PNs, mitigating the occurrence of missed diagnoses and misdiagnoses of LC, enhancing individuals’ quality of life, minimizing the inefficient utilization of medical resources, and facilitating early detection of LC.

Conclusions

The implementation of AI in the detection of PNs has the potential to significantly enhance the accuracy and efficacy of early LC identification, as well as provide more accurate data for clinical decision-making. At the same time, AI technology has the potential to alleviate the burden on medical professionals, enhance the efficacy of lesion identification, and mitigate the occurrence of misdiagnoses. The utilization of AI in the process of identifying PNs has demonstrated a notable degree of accuracy, particularly in the screening and differentiation of benign and malignant cases. Nevertheless, it exhibits superior performance compared to physician interpretation when it comes to distinguishing between benign and malignant lung nodules. Hence, the integration of AI-assisted diagnostics with the expertise of physicians has the potential to enhance the overall efficacy of diagnosing and treating PNs. The future development of AI technology is expected to lead to enhanced precision in the analysis of clinical data, hence improving the accuracy of diagnosis and therapy for patients with PNs.

Supplemental Information

Supplemental Information 1 Raw Data

Click here for additional data file.

Additional Information and Declarations

Competing Interests

Author Contributions

Human Ethics

Data Availability

The authors declare there are no competing interests.

Wei Fan conceived and designed the experiments, performed the experiments, analyzed the data, prepared figures and/or tables, authored or reviewed drafts of the article, and approved the final draft.

Huitong Liu conceived and designed the experiments, authored or reviewed drafts of the article, and approved the final draft.

Yan Zhang conceived and designed the experiments, analyzed the data, prepared figures and/or tables, and approved the final draft.

Xiaolong Chen performed the experiments, authored or reviewed drafts of the article, and approved the final draft.

Minggang Huang performed the experiments, analyzed the data, prepared figures and/or tables, and approved the final draft.

Bingqiang Xu conceived and designed the experiments, analyzed the data, prepared figures and/or tables, authored or reviewed drafts of the article, and approved the final draft.

The following information was supplied relating to ethical approvals (i.e., approving body and any reference numbers):

All samples obtained in this study were approved by the ethics committee of the Shaanxi Provincial People’s Hospital and abided by the ethical guidelines of the Declaration of Helsinki.

The following information was supplied regarding data availability:

The raw data is available in the Supplemental File.

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
