# Peer review of "Diagnostic value of artificial intelligence based on computed tomography (CT) density in benign and malignant pulmonary nodules: a retrospective investigation"

_PeerJ, doi:10.7717/peerj.16577_

## Round 0.1 · original submission · Major Revisions

Please respond and make appropriate revisions based on the reviewers' suggestions and my comments (below). This will greatly improve the quality of the manuscript.

My comments:
1. The vast majority of the results of this study are not related to antioxidants/phytochemicals, and the title, abstract, and conclusion sections of the paper need to be adjusted and revised accordingly.
2. The Discussion could be enhanced by addressing the implications of the study's findings in the context of current literature on AI-assisted diagnosis in lung cancer research. Discussing potential limitations and biases, as well as avenues for future research, would contribute to a more well-rounded conclusion.
3. However, the presentation of AUC (Table 6) could be improved by presenting the data in a more organized manner, such as in graphs, for easy comparison.
4. It would be beneficial to include a brief rationale for the importance of accurate PN diagnosis in the context of lung cancer and the limitations of current diagnostic methods. This would provide readers with a clearer understanding of the study's significance.
5. The discussion of AI could be more specific about how AI helps with detecting nodules.
6. Acknowledge potential limitations of AI, such as its reliance on training data and potential challenges in handling complex cases.
7. Several sentences are quite long and complex, leading to a lack of clarity and coherence.

**Language Note:** The review process has identified that the English language must be improved. PeerJ can provide language editing services - please contact us at [email protected] for pricing (be sure to provide your manuscript number and title). Alternatively, you should make your own arrangements to improve the language quality and provide details in your response letter. – PeerJ Staff

Reviewer 1 ·

Basic reporting

In this study, authors had explored the diagnosis and treatment value of artificial intelligence based on CT density diagnosis combined with antioxidants/phytochemicals treatment in benign and malignant pulmonary nodules. This is an interesting study. Here, I just I give some suggestions to improve the manuscript.
1. The structure of the paper is not complete, please supplement some relevant CT images.
2. It is suggested that the title be changed to “Diagnosis and Treatment Value of Artificial Intelligence Based on CT Density Diagnosis Combined with Antioxidants/Phytochemicals Treatment in Benign and Malignant Pulmonary Nodules: A retrospective investigation”.
3. In the discussion section, authors should discuss the advantages and limitations of their study.
4. Please attach the cited literature for the method section.

Experimental design

1. The format of the paper is not standard, notes, punctuation, paragraphs (in many places, the lack of the basic specification of two spaces) and other aspects, it appears to be very imprecise.
2. The author needs to mention the study of a specific population in the object of study.

Validity of the findings

1. Literature collection of this study is weak, in fact, a lot of research in this area has been carried out, but this paper has not reflected it, resulting in the weak foundation of the paper.
2. The concepts of " pulmonary calcification " and " pulmonary nodule " are different. In this paper, there is a situation of stealing concepts and avoiding confusion.
3. The argument is lack of logic, for example, the third part of the paper has nothing to do with the problems to be studied, and the countermeasures proposed are lack of pertinence.

Additional comments

No.

Reviewer 2 ·

Basic reporting

Good.

Experimental design

The experimental design is reasonable and meets the requirements of the journal. I have no other comments.

Validity of the findings

The findings are basically valid and the results were supported by sufficient data and analysis.

Additional comments

Here are a few suggestions to improve the paragraph:
1. It is noticed that this manuscript requests thoughtful checking by somebody with proficiency English.
2. Providing more background and explanation in the introduction would make the introduction benefit from elucidating why the research is essential and what the research hopes to complete.
3. The word "difference is significant" appears in the article, and whether the difference is significant or not should be seen only after the data has been tested.
4. Clarify the significance of the observations and trials used in the Method section.
5. Evade prolonged sentences and paragraphs for easy legibility.
6. Does the conclusion reasonably answer the questions raised by the author in the introduction?
7. I strongly suggest that the author add some representative CT images to illustrate the advantages of AI in assisting manual examination of pulmonary nodules.

Reviewer 3 ·

Basic reporting

This paper provides an overview of the diagnosis and treatment value of artificial intelligence based on CT density diagnosis combined with antioxidants/phytochemicals treatment in benign and malignant pulmonary nodules. The basic reporting of this manuscript meets the requirements of the journal.

Experimental design

No comment.

Validity of the findings

No comment.

Additional comments

1. Please define in the manuscript why CT diagnosis combined with Antioxidants/Phytochemicals treatment in this study?
2. Provide a clear conclusion that summarizes the key findings of each result.
3. Whether it is appropriate to study the "method" of the main question (perhaps more than one)?
4. Please illustrate the research prospects of this manuscript at the end of the discussion section.
5. Must add 1-2 figures in the draft.
6. Add more citations
7. Add 3-4 paragraphs in the discussion section.
8. Give short summary of the tables data in the table caption.

---

## Round 0.2 · Minor Revisions

Issues that need to be revised:

1. The title should be revised to "Diagnostic Value of Artificial Intelligence Based on CT Density in Benign and Malignant Pulmonary Nodules: A Retrospective Investigation", because this study does not appear to evaluate the role of AI in the treatment of pulmonary nodules.

2. Similarly, in the Abstract, "increase the precision in early detection and therapy of lung carcinoma" could be also misleading. Please delete "and therapy".

3. The legend for Figure 1 was incomplete. Figures 1C and 1D represent malignant nodule? Please clarify it. [diagnosed as invasive ade] means what? Careful proofreading is required.

**Language Note:** The Academic Editor has identified that the English language must be improved. PeerJ can provide language editing services - please contact us at [email protected] for pricing (be sure to provide your manuscript number and title). Alternatively, you should make your own arrangements to improve the language quality and provide details in your response letter. – PeerJ Staff

Reviewer 1 ·

Basic reporting

no comment

Experimental design

no comment

Validity of the findings

no comment

Additional comments

The author has made detailed modifications and improvements to the abstract, methods, results, and experimental design of the article in response to my review comments. I believe that it has met the requirements of the magazine and can be published.

Reviewer 2 ·

Basic reporting

After modification, the structure of the article is clear and clear, and professional English is used throughout the entire process. Very good.

Experimental design

After modifications to the experimental design section, the research questions have become clearly defined, relevant, and meaningful, with sufficient details and information to replicate the described methods.

Validity of the findings

The author has provided all basic data; They are robust, statistically reliable, and controllable.
After modifying the results section, the conclusion statement is sufficient and relevant to the original research question.

Additional comments

I have no further opinions and support accepting this article.

Reviewer 3 ·

Basic reporting

Nice modification. I have no other objections to the article.

Experimental design

Nice modification. I have no other opinions on the experimental design part of the article.

Validity of the findings

Nice modification. I have no further comments on the results section of the article.

Additional comments

No comment. Agree that the article is accepted.

---

## Round 0.3 · accepted · Accept

My concerns were well addressed. I think this revised version could be considered for publication in this journal.